# Designing Physical-World Universal Attacks on Vision Transformers

**Mingzhen Shao**
Kahlert School of Computing
University of Utah
`shao@cs.utah.edu`

## Abstract

Recent studies have highlighted the vulnerability of Vision Transformers (ViTs) to adversarial attacks. However, existing attack methods often overlook the differences between ViTs and CNNs, resulting in difficulties when transitioning attacks from the digital to the physical world. In this work, we introduce a novel adversarial patch generating method, presenting the first physical-world universal attack for ViTs (G-Patch). Unlike previous methods, our approach decouples the relationship between attacker location and ViT patches, enabling the model to design attacks that can occur at random locations in the physical world. To provide a capable learning ability for this more complex situation, we employ a sub-network to craft potential attackers. Our ablation study demonstrates that the previous direct optimization method fails to provide a reliable attack when considering random locations. Our synthetic tests simulate various types of physical-world noise, with G-Patch achieving a targeted attack success rate (ASR) of over 90%, while other approaches exhibit a negligible ASR of less than 10%. Additionally, a black-box attack is designed to demonstrate G-Patch's transferability across different models. A series of challenging physical-world experiments further underscore its robustness in practical deployments.

## 1 Introduction

Vision transformers (ViTs) have garnered significant attention due to their impressive performance and their ability to surpass convolutional neural networks (CNNs) in various domains Dosovitskiy et al. (2020); Chen et al. (2021a;b); Graham et al. (2021); Han et al. (2021); Liu et al. (2021); Touvron et al. (2021); Xiao et al. (2021). This remarkable performance has spurred interest in examining the robustness of ViTs, particularly considering the well-known vulnerability of CNNs to adversarial attacks Bhojanapalli et al. (2021); Qin et al. (2022); Salman et al. (2022); Shi et al. (2022).

Like attacks on CNNs, adversarial attacks on ViTs can be classified as either digital or physical, based on the deployment environment. Digital attacks usually can be very efficient, sometimes only modifying a few values of the input images Fu et al. (2022); Gu et al. (2022); Wei et al. (2022); Wang et al. (2022). However, a critical limitation of these methods lies in the requirement to access digital source images.

The realm of physical attacks offers a more realistic environment for real-world deployment scenarios, where attacks are limited to modifying the objects interacting with the system. Physical attacks have demonstrated great success in CNNs, attributed to their reasonable prerequisites and robustness Kurakin et al. (2018); Athalye et al. (2018); Eykholt et al. (2018); Brown et al. (2017); Hu et al. (2021); Zhang et al. (2019); Thys et al. (2019); Wu et al. (2020b).

However, physical attacks on ViTs face significant challenges due to their distinct architecture. Unlike CNNs, ViTs represent an input image as a sequence of image patches. Previous research has demonstrated ViTs' resilience against perturbations when the entire input image is disturbed Bhojanapalli et al. (2021); Aldahdooh et al. (2021); Bai et al. (2021); Shao et al. (2021); Mahmood et al. (2021). Additionally, existing patch-based attacks Fu et al. (2022); Gu et al. (2022) have been shown to be highly sensitive to the position of the attack samples. Gu Gu et al. (2022) demonstrated

that even a minor one-pixel misalignment results in a 70% performance loss. The attack samples generated through these approaches cannot be effectively deployed in the physical world, as there is no guarantee of seamless replacement of specific input image patches.

The properties of existing attacks on ViTs are summarized in Table 1. The 'universal' indicates the approach's ability to launch an attack without prior knowledge of the scene's elements.

Table 1: Summary of existing attacks and our G-Patch.

| Attack | Digital | Physical | Universal |
|---|---|---|---|
| Lavan Karmon et al. (2018) | ✓ | × | ✓ |
| TransferAdv Wei et al. (2022) | ✓ | × | × |
| ATA Wang et al. (2022) | ✓ | × | × |
| PatchFool Fu et al. (2022) | ✓ | × | × |
| G-Patch (ours) | ✓ | ✓ | ✓ |

To enable a universal attack in the physical world, we propose the G-Patch generating model, for the **first time** decouple the relationship between attacker location and ViT patches. We employ a deployer structure, commonly used in designing attacks for CNNs, to provide random locations for the potential patches. However, as shown in our ablation study, simply providing a random location during training does not yield a workable attack. The other challenge lies in the learning ability of the attacker generating model. Therefore, we utilize a five-layer sub-network (generator) to create potential G-Patches instead of directly optimizing the patches.

A synthetic test is established to simulate random noise and slight deformations commonly encountered in the physical world. In this synthetic test, the G-Patch achieves a targeted attack success rate (ASR) of over 90% with a small size ($\sim 10\%$ of the source image), while other approaches exhibit a negligible ASR (less than 10%). Through ablation studies, we show the significant impact of the learning boosted generator and random position deployer on achieving a substantial performance improvement. Furthermore, we design a black-box attack to show the transferability of our G-Patch.

In the physical world experiments, we introduce additional challenging factors such as varying/uneven lighting conditions, different capture angles, and placing on non-flat surfaces. While the ASR may show a decrease compared to the synthetic tests, the robust performance of the G-Patch in these experiments underscores its efficacy as a powerful attack in real-world scenarios.

Our contributions can be summarized as follows:

- We introduce G-Patch, marking the first instance of launching physical-world universal attacks on ViTs.
- We demonstrate that the learning ability required to generate a random location attacker for ViTs is significantly greater than that for CNNs.
- We introduce a new synthetic test in the digital domain to simulate the rotation and shiftiness encountered in the physical world, enabling a more realistic evaluation of the ASR of attacks.

## 2 RELATED WORK

### 2.1 VISION TRANSFORMER

The transformer was first introduced by Vaswani Vaswani et al. (2017) for natural language processing (NLP) tasks. Following the success in NLP, Dosovitskiy Dosovitskiy et al. (2020) proposed the vision transformer that leveraged non-overlapping patches as tokens input to a similar attention based architecture. Since then, numerous models have been proposed to alleviate training challenges or enhance the performance of vision transformer models. Touvron Touvron et al. (2021) introduced a teacher-student strategy in their DeiT models that dramatically reduced the pre-training request. Liu Liu et al. (2021) proposed the SWIN transformer using the shifted windowing scheme that achieves greater efficiency by limiting self-attention computation to non-overlapping local windows

while also allowing for cross-window connection. As the vision transformer continues to advance, achieving state-of-the-art performance and becoming increasingly accessible for pre-training Zhang et al. (2021); Tu et al. (2022); Dong et al. (2022); Zhai et al. (2022); Yao et al. (2023), it has seen widespread adoption in diverse visual tasks, including video processing Arnab et al. (2021); Liu et al. (2022), dense prediction Ranftl et al. (2021) zero-shot classification Radford et al. (2021), captioning Li et al. (2022), and image generation Rombach et al. (2022).

### 2.2 ADVERSARIAL ATTACK

#### 2.2.1 DIGITAL ADVERSARIAL ATTACK

The first digital adversarial attack for computer vision tasks was introduced by Szegedy Szegedy et al. (2013). Since their seminal work, numerous researchers have devised increasingly efficient techniques for generating adversarial attacks Moosavi-Dezfooli et al. (2017); Athalye et al. (2018); Huang et al. (2019); Karmon et al. (2018); Brown et al. (2017). Classic digital attack methods include gradient-based methods (, FGSM Goodfellow et al. (2014), PGD Madry et al. (2017), SGD Wu et al. (2020a)), optimization-based methods (, C&W Carlini & Wagner (2017), ZOO Chen et al. (2017)), and GAN based methods (, AC-GAN Song et al. (2018) AdvGAN Xiao et al. (2018), PS-GAN Liu et al. (2019), GDPA Li & Ji (2021)). One critical limitation of these methods lies in the necessity of modifying the digital values of input images, which proves challenging to replicate when deploying in real-world scenarios.

#### 2.2.2 PHYSICAL ADVERSARIAL ATTACK

The primary advancement in physical adversarial attacks is restricting attackers to manipulate only the environment or objects directly interacting with the system. Kurakin Kurakin et al. (2018) proposed the first physical adversarial attack model by printing digital adversarial examples onto paper. They found that a significant portion of the printed adversarial examples deceived the image classifier. Athalye Athalye et al. (2018) improved on this work by creating adversarial objects that remain effective even when viewed from different angles. They achieved this by modeling small-scale transformations synthetically when generating adversarial perturbations. Eykholt Eykholt et al. (2018) modeled image transformations both synthetically and physically. Their work demonstrated that relying solely on synthetic transformations will reduce attack robustness. However, all these approaches necessitate tailoring the design of each attack to a specific target image, requiring the creation of a new generation for each new attack (non-universal). This process proves inefficient and, in some cases, impractical for real-world deployment.

The first universal attack approach was proposed by Brown Brown et al. (2017). They used gradient-based optimization to iteratively update the pixel values of a patch to find the optimal values that can cause the victim model to misclassify the object. This patch can initiate an attack without knowing the other items within the scene, enabling attackers to craft physical-world attacks easily. Since then, many studies have followed the same strategy to develop patches for physical-world attacks aimed at deceiving classifiers or object detectors Hu et al. (2021); Zhang et al. (2019); Thys et al. (2019); Wu et al. (2020b).

### 2.3 ADVERSARIAL ATTACK ON VISION TRANSFORMER

Shortly after the introduction of the vision transformer, Wei Wei et al. (2022) generated a transferable adversarial example by skipping the gradients of attention during backpropagation. Wang Wang et al. (2022) proposed an Architecture-oriented Transferable Attacking (ATA) framework to generate transferable adversarial examples by activating the uncertain attention and perturbing the sensitive embedding.

In contrast to approaches that perturb the entire image with adversarial perturbations (non-universal attack), some research focuses on patch-based attacks (universal attack). Fu Fu et al. (2022) explored the vulnerability of vision transformers to adversarial patch attacks and found that vision transformers are more susceptible to such attacks than CNNs. Additionally, Gu Gu et al. (2022) further showed that whereas vision transformers are generally resilient to patch-based natural attacks, they are more vulnerable to adversarial patch attacks when compared to comparable CNNs. Chen Chen et al. (2023) introduced a decision-based approach for crafting attack patches tailored

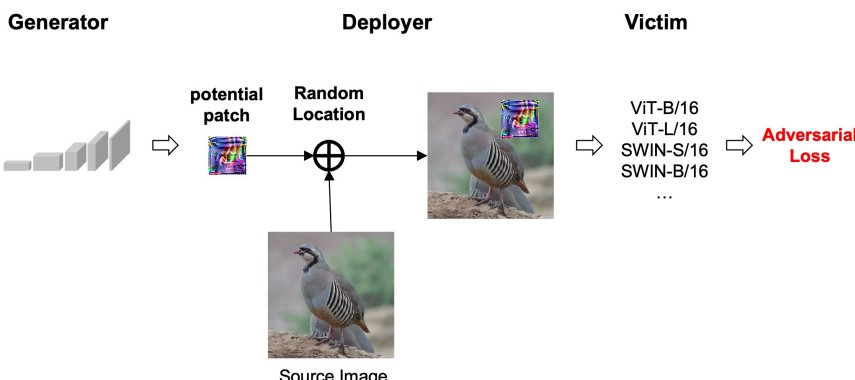

Figure 1: Overview of the G-Patch generating model. Unlike GAN-based models, the victim is not modified during training.

to vision transformers. Unlike learning-based approaches, their method requires patch regeneration for each target image, rendering it a non-universal attack.

A significant drawback of patch-based approaches is their strict requirement for the precise location of the attack patch. Gu . Gu et al. (2022) illustrated that even a minor misalignment of a single pixel could result in a drastic performance decrease of over 70%. This constraint impedes deployment in the physical world, where ensuring the accurate location is not guaranteed.

## 3 G-PATCH GENERATING MODEL

The overview of our G-Patch generating model is illustrated in Figure 1. Our approach involves training a sub-network (Generator) to create adversarial patches from random inputs. Deployer is designed to simulate random locating that an adversarial patch will encounter, to compel the generator to create robust patches for physical-world deployment. The outputs of the deployer are then fed to the victim to calculate the adversarial loss, which is subsequently used to optimize the generator.

### 3.1 GENERATOR

The generator is a sub-network consisting of five convolutional layers, each accompanied by batch normalization and ReLU activation layers.

The last convolutional layer is followed by a threshold layer instead of the batch normalization and ReLU layers. This threshold layer ensures that the output values of the generator are limited to a specific range. The threshold layer is defined as follows:

$$Th(x) = k * tanh(x) + k, \quad 0 < k \le 0.5 \tag{1}$$

where $k$ is a hyperparameter to adjust the range of the output and $tanh(x)$ applies the hyperbolic tangent function element-wise. We add $k$ here to ensure that all values in the output remain in the range [0,1].

By default, we employ $k = 0.5$ to normalize the patch range to 1. A smaller value for $k$ can be employed to limit the value range for the G-Patch, reducing the neon color appearance in the generated patch.

Convolutional layers require different kernel sizes and strides to generate G-Patches with varying sizes. Let $Ck(s)$ denote a Convolution-Norm-ReLU layer with a kernel size of $k$ and stride $s$, and $CTk(s)$ denote the final Convolution-Threshold layer with a kernel size of $k$ and stride $s$. The architecture of the generator, which produces G-Patch with different sizes, is illustrated as follows:
**G-Patch(64):** $C4(1) - C4(2) - C4(2) - C4(2) - CT4(2)$
**G-Patch(80):** $C6(1) - C5(1) - C4(2) - C4(2) - CT4(2)$

**G-Patch(96):** $C7(1) - C6(1) - C4(2) - C4(2) - CT4(2)$

## 3.2 DEPLOYER

In physical-world deployments, the adversarial patch can be positioned anywhere within the scene. Previous methods, which replace input patches for ViTs, are too idealized and not suitable for real-world scenarios. To address this, the deployer places the potential G-Patch at a random position within the source image $I$, avoiding placement near the borders. Additionally, a rotation of $[0, \pm90, 180]$ is applied to the patch to enhance its robustness. The resulting image with the rotated G-Patch is denoted as $\hat{I}$.

## 3.3 ADVERSARIAL LOSS

The victim is employed to classify the modified image $\hat{I}$ and compute an adversarial loss, similar to other approaches. The victim can be a single target model for generating a specific white-box attack or a combination of various target models for generating a black-box attack.

For a targeted attack, the loss for each target model is:

$$loss_n = log(softmax(Pr_n(\hat{y}|\hat{I}))) \tag{2}$$

where the $\hat{y}$ is the target class, $Pr_n$ is the prediction of the $n^{\text{th}}$ target model with respect to class $\hat{y}$.

The final adversarial loss of our network can be formed as follows:

$$L_{adv} = \frac{1}{n}(loss_1 + loss_2 + ... + loss_n) \tag{3}$$

## 4 EXPERIMENTAL RESULTS AND ANALYSIS

In this section, we started with the experimental setup used in our study. We then present G-Patches generated by our proposed model, comparing their performance with other state-of-the-art adversarial attacks such as TransferAdv Wei et al. (2022), ATA Wang et al. (2022), and PatchFool Fu et al. (2022) in synthetic tests. Through a series of ablation studies, we investigate the relationship between the ASR and the attacker generating model parameters. and demonstrate the efficacy of the random location model in addressing challenges encountered in physical-world deployment. Additionally, we conduct a robustness analysis of G-Patches in black-box attacks. Finally, we validate the practical applicability of the G-Patch by deploying it in real-world scenarios, showing its effectiveness in complex physical environments.

## 4.1 EXPERIMENTAL SETUP

In our experiments, we use the weights and shared models from the *Pytorch Image models* repository Wightman (2019), which have been trained on the ImageNet1K dataset.

Similar to the pre-existing works, the effectiveness of the G-Patch is evaluated using white-box attack settings, where the G-Patch is trained and tested on the same models. Specifically, we choose the ViT and SWIN transformer as the primary victim network architectures. These two networks exemplify crucial differences in patch handling within vision transformers: while the ViT employs fixed, non-overlapping patches, the SWIN transformer integrates varying patch sizes with shifts.

We optimize our generator using the adversarial loss $L_{adv}$ with the Adam optimizer ($lr = 0.001$) and conduct training for each configuration over 80 epochs, selecting the patch that achieves the highest performance as the final output patch. Additionally, input images are standardized to dimensions of 224x224, with pixel values normalized to the range of [0,1] for consistency.

The **synthetic tests** involve applying rotations of $\pm5°$ and shifts of $\pm14$ pixels to the modified images. Afterward, the updated images are resized to 224x224 pixels, with blank areas filled in using black color. Subsequently, these updated images are fed into the target model to calculate the ASR. Figure 2 provides a visual representation of the images used in synthetic tests for various approaches.

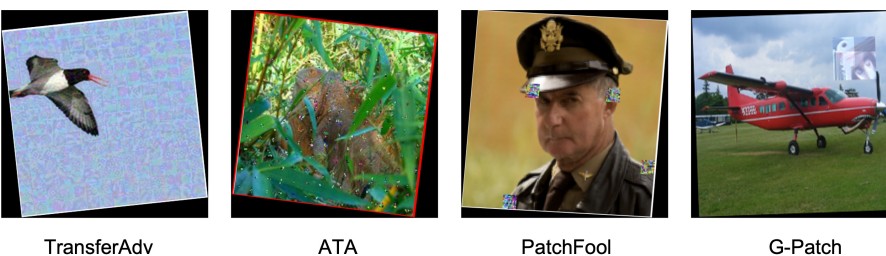

TransferAdv     ATA     PatchFool     G-Patch

Figure 2: Images across different approaches in synthetic tests.

In order to assess the patch's efficacy in real-world scenarios, we employ an HP laser printer to reproduce adversarial patches on A4 paper. Subsequently, we place the printed patch adjacent to the target object and capture images with a Google Pixel 6a smartphone. The captured images are also standardized to dimensions of 224x224, with pixel values normalized to the range of [0,1]. This physical-world evaluation introduces diverse real-world elements, including varying/uneven lighting conditions, different capture angles, and even placement on non-flat surface.

## 4.2 PERFORMANCE IN SYNTHETIC TEST

Table 2: The ASR (%) results across various ViTs in synthetic tests. *(G-Patch(64) - 8% input image, G-Patch(80) - 12% input image, G-Patch(96) - 18% input image.)*

| Methods | ViT-B | ViT-L | SWIN-S | SWIN-B | DeiT-B |
|---------|-------|-------|--------|--------|--------|
| TransferAdv | 0.4 | 0 | 2.4 | 1.3 | 0.6 |
| ATA | 0.7 | 0.5 | 2.4 | 1.9 | 1.8 |
| PatchFool | 5.7 | 4.4 | 7.4 | 9.3 | 4.6 |
| G-Patch(64) | 75.4 | 63.3 | 93.7 | 92.8 | 96.6 |
| G-Patch(80) | 96.1 | 87.7 | 98.5 | 98.3 | 98.9 |
| G-Patch(96) | 98.7 | 97.6 | 99.9 | 99.8 | 99.9 |

In Table 2, the G-Patches demonstrate significantly greater effectiveness in attacking ViTs. Earlier approaches, in our synthetic tests, exhibit negligible ASR, *in stark contrast to the over 90% ASR reported for the same models without considering rotations and shifts*. This dramatic drop in ASR clearly demonstrates the unsuitability of these earlier methods for physical-world deployments.

## 4.3 ABLATION STUDY

In this section, we highlight that both the generator and deployer structures are essential for generating physical-world attacks on ViTs.

### 4.3.1 ENHANCED LEARNING ABILITY REQUIRED FOR ViT ATTACKS

To analyze the impact of the generator, we compare patch generating methods with three different learning abilities: our five-layer generator, a simplified three-layer generator, and the previous direct optimization used in Fu et al. (2022); Qin et al. (2022), with the learnable parameters of the different methods being approximately 3M, 1M, and 0.01M, respectively. All of the methods include a random location deployer during training.

The direct optimization method eliminates the generator, inputting a random potential patch directly into the deployer. Inspired by previous works Fu et al. (2022); Gu et al. (2022), this approach allows the adversarial loss to directly optimize the potential patch. The simplified three-layer generator shares the similar structure as our five-layer generator but uses fewer layers.

The ASR of various generating methods with different learning abilities is shown in Table 3. We observe a significant improvement in ASR by enhancing the learning ability of the generating model.

This indicates that the limited learning ability of the previous direct optimization method is a key obstacle in generating effective physical world attacks on ViTs.

Table 3: ASR (%) for methods with different learning abilities.

| Learnable Parameters | 12288 | 1.2 M | 2.8 M |
|---|---|---|---|
| ViT-B | 0.6 | 66.5 | 98.7 |
| SWIN-B | 1.4 | 73.4 | 99.9 |

While direct optimization works well for generating attacks on CNNs, our experiments show it is inadequate for generating attacks on ViTs. This highlights the enhanced learning ability required for ViT attacks, aligning with the claim that ViTs are more robust to adversarial attacks Shao et al. (2021).

### 4.3.2 DECOUPLING ATTACKER'S LOCATION FROM VIT PATCHES

Our deployer structure decouples the attacker's location from ViT patches, allowing attackers to be positioned randomly rather than precisely replacing input patches for ViTs. Here, we conduct experiments training G-Patches without using the deployer during training, while continuing to use the five-layer generator (G-Patch$_{w/o\ D}$).

The synthetic test results are presented in Table 4. Without the random location deployer, the generator encounters only the simplest conditions during training, where the G-Patch precisely replaces some input image patches. Consequently, such G-Patches are ineffective in synthetic tests unless they accidentally align with specific input image patches, as observed in similar studies Gu et al. (2022).

This result demonstrates that the deployer is necessary to enable a physical-world attack for ViTs. Even with a powerful generator, attacks generated without a random location deployer cannot achieve a high ASR in the physical world.

Table 4: The ASR (%) results without Deployer.

| Methods | ViT-B | SWIN-B | DeiT-B |
|---|---|---|---|
| G-Patch$_{w/o\ D}$(64) | 7.6 | 10.6 | 8.4 |
| G-Patch$_{w/o\ D}$(80) | 10.7 | 13.0 | 11.1 |
| G-Patch$_{w/o\ D}$(96) | 12.5 | 14.8 | 11.4 |

Table 5: The ASR (%) results in black-box targeted attacks.

| Model | TransferAdv | ATA | PatchFool | G-Patch$_B$(80) | G-Patch$_B$(96) |
|---|---|---|---|---|---|
| SWIN-B | 0.4 | 0.5 | 1.2 | 50.4 | 76.4 |

### 4.4 BLACK BOX ATTACK

We conducted a black-box attack to demonstrate the transferability of our G-Patch. In this experiment, our victim comprises ViT-B, ViT-L, DeiT-S, DeiT-B, and VGG16, which were used to generate the G-Patch. Subsequently, we employed the SWIN-B model as the black-box target to assess transferability. Additionally, following the methodology outlined in Fu et al. (2022), we integrated a CNN model into the training process to enhance transferability. ASR evaluation was conducted under the same synthetic test settings, and the corresponding results are summarized in Table 5.

We're not surprised to see previous approaches yielding minimal ASR in this black-box test, given their poor performance in the white-box synthetic tests. In contrast, our G-Patch, while experiencing an ASR decrease, demonstrates robust transferability, especially with a larger patch size (G-Patch$_B$(96)). This observation aligns with findings from universal adversarial patches in

CNNs Brown et al. (2017), where larger adversarial patches are required for black-box attacks compared to white-box attacks.

## 4.5 PHYSICAL WORLD ATTACK

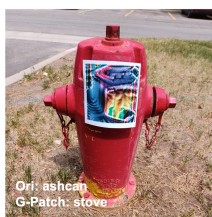 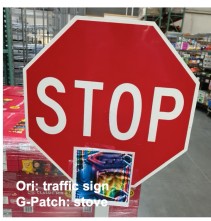 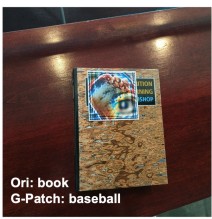 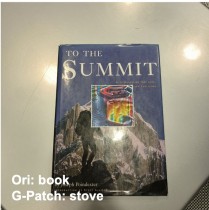

Figure 3: G-Patch attack results in the physical world.

The deployability in physical-world scenarios is a key aspect that contributes to the popularity of adversarial patches over many other attacking methods. Until now, none of the adversarial patches tailored for vision transformers have been successfully deployed in the physical world, primarily due to challenges related to positioning. Although our G-Patches show high ASR in previous synthetic tests, a valid concern remains regarding their robustness in physical-world scenarios. In order to address this concern, we designed several physical-world deployment instances to demonstrate that the proposed attack patch can still function robustly.

Our experiments encompass a broad spectrum of lighting conditions, ranging from 3000K (warm white) to 5500K (daylight), and camera capture angles from straight on to 30 degrees. These images were taken with a regular cellphone camera without any specific constraints. We employed a ViT-B model as the victim model for these tests. Sample images and predictions are provided in Figure 3.

The first image in Figure 3 depicts an extreme condition where the G-Patch is not placed on a flat surface. The second image illustrates the attack under a large capture angle, while the third and fourth images depict the attack under uneven lighting conditions and reflections. The success observed under these challenging conditions underscores the remarkable effectiveness of the G-Patch in executing targeted attacks within complex physical environments.

These results demonstrate the robustness of G-Patch in launching attacks in the complex physical world, addressing practical challenges encountered when designing adversarial patches for vision transformers.

## 5 CONCLUSION

This paper introduces the first physical-world universal attack on vision transformers. Unlike previous approaches, our model decouples the attacker's (G-Patch) location from ViT patches and enhances the learning ability of the attacker generating model. We design a synthetic test to simulate common occurrences such as rotation and shifting in physical-world deployment. Under this more realistic evaluation, our G-Patch significantly outperforms previous approaches. Ablation studies highlight the enhanced learning ability required for ViT attacks and the necessity of both the generator and deployer in creating our G-Patch. A black-box attack underscores the significant transferability of the G-Patch, removing the final obstacle to physical-world deployment. Our physical-world experiments include various challenging conditions, such as varying or uneven lighting conditions, different capture angles, and even placement on non-flat surfaces. The G-Patch effectively bridges the practicality gap between digital and physical realms for adversarial patches on vision transformers, paving the way for expansive future research endeavors.

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
