# OpenReview forum: "Designing Physical-World Universal Attacks on Vision Transformers"
_NeurIPS.cc/2024/Workshop/SafeGenAi — SafeGenAi Poster_

### Official Review · Reviewer_wcvu · 2024-10-08
**DESIGNING PHYSICAL-WORLD UNIVERSAL ATTACKS ON VISION TRANSFORMERS**

**Rating:** 7
**Confidence:** 5

**Review:**

I like this work, here are the pros and concerns:

advantage:
1. Writing is understandable
2. G-Patch uses an ingenious design to create a five-layer sub-network (generator) to create a potential adversarial patch, which decouples the relationship between attacker location and ViT patches, enabling the model to design attacks that can occur at random locations in the physical world.
3. The experiment is solid.

Concerns:
1. There is no more direct basis for why previous work cannot achieve generalization and positional randomness. Supplementing the comparison with previous work is recommended.
2. It is recommended to consider the size of the adversarial patch in the experiment, because as far as the deployment scenario in Section 4.5 is concerned, this attack may be useless due to an excessively large patch size.
3. I am curious whether the attack can be achieved by deploying the patch anywhere. If not, please add a section to discuss its attack boundary. If so, it is recommended to conduct a more systematic experiment. You can select several locations in the victim object and conduct the experiment in sequence, which may be more convincing.

---

### Official Review · Reviewer_acgT · 2024-10-08
**G-Patch: Bridging the Gap Between Digital and Physical Adversarial Attacks on Vision Transformers**

**Rating:** 7
**Confidence:** 4

**Review:**

This paper presents a novel approach for generating adversarial patches to attack Vision Transformer (ViT) models in physical-world settings. The authors introduce "G-Patch", which they claim is the first physical-world universal attack on ViTs. The key innovations are: 1) Decoupling the attacker's location from the ViT input patches using a "deployer" module, and 2) Enhancing the learning ability of the patch generator using a 5-layer convolutional network. The decoupling of location from patches is particularly noteworthy, as it adds significant robustness to the attacks.

The methodology is thoroughly described and evaluated through both synthetic tests and physical-world experiments. The synthetic tests, presented in Table 2, simulate physical-world conditions like rotations and shifts, demonstrating significantly higher attack success rates (ASR) for G-Patch compared to existing digital attacks on ViTs.

The physical-world experiments, while limited in scale, provide compelling evidence of G-Patch's effectiveness under challenging real-world conditions, including various lighting conditions (3000K to 5500K) and camera capture angles (up to 30 degrees).

One of the paper's strengths is addressing the practical challenges of attacking ViTs in physical settings. The authors identify key limitations of existing patch-based attacks on ViTs, particularly their sensitivity to precise positioning. The proposed G-Patch overcomes this by learning to generate robust patches that work across random locations. The physical-world experiments shown in Figure 3 are particularly interesting, demonstrating the attack's effectiveness on non-flat surfaces and under uneven lighting conditions.

However, there are some limitations and areas for clarification. The scale of the physical-world experiments is unclear - the paper only presents results for 4 objects/attacks, and it would be beneficial to know how many objects and patches were tried in total. The decision process for determining patch sizes is not clear to me (besides being 224x224), as the ratio of patch to object seems to vary across examples. Is the distance a factor here? How well did the attacks work from varying distances/angles in the physical world? Additionally, a baseline comparison using random noise patches of the same size would provide context for evaluating G-Patch's performance.

Minor note: The paper would also benefit from highlighting the best results in each row of Table 2 for easier reading.

Despite these limitations, this paper makes a novel contribution to the under-studied yet crucial field of physical adversarial attacks on ViTs. The thorough evaluation and ablation studies provide valuable insights into the challenges of attacking ViTs in practical settings. While there are areas for improvement and clarification, the work opens up promising directions for future research in this domain. The paper would be a good addition to this workshop, spurring discussions on the robustness of ViTs in real-world deployments.

---

### Official Review · Reviewer_23po · 2024-10-10

**Rating:** 5
**Confidence:** 4

**Review:**

### **Summary**
In this paper the authors propose a adversarial patch generating method, for physical-world universal attacks on ViTs(G-Patch) where the relationship between attacker location ViT patches is decoupled. The authors conduct experiments in both digital and physical domains to verify the effectiveness of the method.

Overall this paper lacks ablation studies and experiments for attacks specific for ViT(mentioned in cons) and physical world attacks which is the main direction of the paper and compares against only attacks which have been previously successful digitally. Although it has other experiments this is marginally below acceptance threshold.

### **Pros**

- The experiments conducted demonstrate the effectiveness of G-Patch in achieving high targeted attack success rates in comparison to existing methods
- The authors have also experimented in the physical world have shown transferability from digital setting to physical setting

### **Cons**
- The authors claim that the *learning ability required to generate a random location attacker for
ViTs is significantly greater than that for CNNs* but this hasn't been shown in the experiments
- The authors claim this is a physical-world universal attack but haven't compared their model against baselines which attack in the physical world such as [1][2]
- The attack which is around 10% is quite large and conspicuous in comparison to baseline methods. The patches also seem very big for the outdoor scenario.
- No comparison against existing work on attack on ViT like DevPatch[2]

[1] Robust physical-world attacks on machine learning models \
[2] Making an Invisibility Cloak: Real World Adversarial Attacks on Object Detectors \
[3] Query-efficient decision based black-box patch attack

#### **Misc.**
- The Authors double reference the lead author names throughout the paper. Suggest them to fix this typo.